# Quantification of Furosine (N^ε^-(2-Furoylmethyl)-l-lysine) in Different Parts of Velvet Antler with Various Processing Methods and Factors Affecting Its Formation

**DOI:** 10.3390/molecules24071255

**Published:** 2019-03-31

**Authors:** Rui-ze Gong, Yan-hua Wang, Kun Gao, Lei Zhang, Chang Liu, Ze-shuai Wang, Yu-fang Wang, Yin-shi Sun

**Affiliations:** 1Institute of Special Animal and Plant Sciences, Chinese Academy of Agricultural Sciences, Changchun 130112, China; 82101172456@caas.cn (R.-z.G.); yhwangsdlc@126.com (Y.-h.W.); 13356954028@163.com (K.G.); leizhang0102@163.com (L.Z.); liuchang@caas.com (C.L.); 13091716585@163.com (Z.-s.W.); wangyufang_jl@163.com (Y.-f.W.); 2College of Chinese Material Medicine, Jilin Agricultural University, Changchun 130118, China

**Keywords:** affecting factors, amadori compound, furosine, Maillard reaction, velvet antler processing

## Abstract

Furosine (N^ε^-(2-furoylmethyl)-l-lysine) is formed during the early stages of the Maillard reaction from a lysine Amadori compound and is frequently used as a marker of reaction progress. Furosine is toxic, with significant effects on animal livers, kidneys, and other organs. However, reports on the formation of furosine in processed velvet antler are scarce. In this study, we have quantified the furosine content in processed velvet antler by using UPLC-MS/MS. The furosine contents of velvet antler after freeze-drying, boiling, and processing without and with blood were 148.51–193.93, 168.10–241.22, 60.29–80.33, and 115.18–138.99 mg/kg protein, respectively. The factors affecting furosine formation in processed velvet antler, including reducing sugars, proteins, amino acids, and process temperature, are discussed herein. Proteins, amino acids, and reducing sugars are substrates for the Maillard reaction and most significantly influence the furosine content in the processed velvet antler. High temperatures induce the production of furosine in boiled velvet antler but not in the freeze-dried samples, whereas more furosine is produced in velvet antler processed with blood, which is rich in proteins, amino acids, and reducing sugars, than in the samples processed without blood. Finally, wax slices rich in proteins, amino acids, and reducing sugars produced more furosine than the other parts of the velvet antler. These data provide a reference for guiding the production of low-furosine velvet antler and can be used to estimate the consumer intake of furosine from processed velvet antler.

## 1. Introduction

Velvet antler is an important ingredient in traditional Chinese medicine that has been used for thousands of years in China, Korea, and Southeast Asian countries [1,2,3]. Velvet antler contains anti-oxidants and other compounds associated with immunity, anti-osteoporosis, and other pharmacological effects [4,5,6]. Fresh velvet antler is rich in blood, amino acids, and proteins, and is highly susceptible to spoilage if it is not processed promptly. Based on current processing methods and consumption patterns, velvet antler is mainly boiled or freeze-dried. It can be processed with or without blood and separated into wax, powder, gauze, and bone slices by segmentation. The various velvet antler processing techniques have different impacts on the bioactive components and pharmacological activities [4,5,6]. Therefore, the processing conditions are crucial for the resulting composition of the velvet antler.

Velvet antler is rich in amino- and carbonyl-containing compounds, which can produce advanced glycation end products (AGEs) via the Maillard reaction during processing [7]. Some AGEs have been associated with a variety of diseases, including diabetes, Alzheimer’s disease, atherosclerosis, renal dysfunction, and aging [8,9,10,11,12,13,14,15,16,17]. Our group has reported the content of N^ε^-(carboxymethyl) lysine (CML) and N^ε^-(carboxyethyl) lysine (CEL) in different parts of velvet antler, processed by using different methods [18]. The contents of CML and CEL in boiled velvet antler and samples processed with blood were significantly higher than those in freeze-dried velvet antler and samples processed without blood, indicating that different processing methods can significantly affect the degree to which the Maillard reaction occurs. However, pre-treatment methods for CML and CEL in processed velvet antler are cumbersome and the compounds have no UV absorption or fluorescence characteristics, thereby necessitating LC-MS analysis [19,20,21].

Furosine (N^ε^-(2-furoylmethyl)-l-lysine or N6-[2-(2-furanyl)-2-oxoethyl]-l-lysine, C_12_H_18_N_2_O_4_, Mw 254.12, FML) is a product of lysine Amadori compounds, its formation pathway, which is a part of the early stages of the Maillard reaction, as shown in Figure 1 [22]. Furosine can further react to form AGEs and can be used as a marker of Maillard reaction progress [23]. Li et al. [24] showed that furosine has strong toxic effects on animal livers, kidneys, and other organs. High doses of furosine cause adverse effects on health by inducing cell apoptosis and activation of inflammatory necrosis. Furosine has maximum UV absorption at 280 nm and is easily detected. However, furosine is a trace substance in foods and drugs. As LC-MS has better sensitivity and selectivity than UV measurements, LC-MS is now usually used to detect furosine in foods and drugs [19]. Furosine has been used to evaluate the shelf life and freshness of foods, including milk and honey [25,26]. However, data regarding furosine in processed velvet antler have not been reported to date. The furosine content in processed velvet antler is significantly influenced by the matrix and processing conditions [27,28]. Therefore, information regarding the furosine content in processed velvet antler is required to evaluate the quality of processed velvet antler.

This study was performed to determine the furosine content in processed velvet antler by UPLC-MS/MS and to explore the factors affecting the content by measuring the contents of furosine, amino compounds (proteins and amino acids), and carbonyl compounds (reducing sugars) in different parts of the velvet antler, processed using various methods. In addition, the conditions affecting the production of furosine in the Maillard reaction during processing were analyzed. This study provides a solid theoretical basis for improving the processing technology of velvet antler and for controlling the degree of processing and production of furosine. This study also provides a reference for consumers to control their furosine intake from processed velvet antler.

## 2. Results and Discussion

### 2.1. Sample Pre-Treatment and Chromatography Conditions

Preparation of the velvet antler samples consisted of processing, segmenting, grinding, hydrolysis, and solid-phase extraction (SPE). Unlike the sample pre-treatment for CML and CEL, the pre-treatment method for detecting furosine does not involve defatting or reduction and is relatively simple. The samples were subjected to SPE by using a C18 Sep-Pak^®^ cartridge (Sepax technology, Cork, Ireland; 500 mg, 6 mL) to remove impurities.

Furosine is a highly polar compound and is not well retained by most reversed-phase columns. Researchers have usually analyzed furosine by using a C18 column with nonafluoropentanoic acid (NFPA) and trifluoroacetic acid (TFA) as eluents to improve peak patterns and reduce tailing. However, NFPA and TFA can result in poor mass spectra and reduce the service life of the instrument [29]. To avoid using NFPA and TFA, we developed a UPLC-MS/MS method to separate furosine with an Acquity HSS T3 UPLC column. The HSS T3 column is a reverse-reverse column with excellent retention of highly polar compounds, relative to other commonly used columns. It relies on ion exchange and hydrophobic interactions between the stationary phase and furosine to achieve separation. In comparison with the HILIC column, the HSS T3 column stationary phase is compatible with 100% water and has a wider elution range. The elution effects of methanol and acetonitrile were assessed by using acetonitrile/water (80:20 *v*/*v*) and methanol/water (80:20 *v*/*v*) mixtures as mobile phases, with a flow rate of 0.3 mL/min. In comparison with the chromatograms obtained by using a methanol mobile phase, UPLC-MS chromatograms obtained by using water and acetonitrile as eluents yielded better spectrum peak symmetries and fewer miscellaneous peaks.

The UPLC-MS chromatograms of the furosine standard and a sample of powder slices of boiled velvet antler are shown in Figure 2a,b. In the Appendix A, we provide a UV determination of furosine standards and powder slices of boiled velvet antler samples. The same retention time, UV measurement, and total ion chromatograms for furosine in the processed velvet antler samples were consistent with those of the furosine standard and no peak interference was observed.

### 2.2. Method Validation

The developed method was validated by assessing the furosine content in processed velvet antler samples and considering the resulting selectivity, linearity, precision, and accuracy. Figure 2c shows the fragmentation pattern of furosine, with three major product ions at *m*/*z* 84, 130, and 192, with the most intense peak at *m*/*z* 130. The three product ions were used for quantitation in multiple reaction monitoring (MRM) mode. Figure 2e shows the assignment of the furosine fragmentation pattern.

The correlation coefficient (R^2^) of furosine was ˃0.9998 and the linear range (20–3500 ng/mL) was sufficiently wide to assess the furosine content in the processed velvet antler samples. The reference solution was diluted stepwise with ultrapure water. The limit of detection (LOD) and limit of quantitation (LOQ) were defined as the concentrations at which the signal-to-noise ratios of the furosine peak were 3 and 10, respectively. The LOD and LOQ values for furosine were 1.9 ng/g and 5.7 ng/g, respectively.

The processed velvet antler samples were extracted in triplicate and analyzed by using the developed UPLC-MS/MS method. The relative standard deviations of the intra-day and inter-day precision for furosine were 3.12 and 4.28%, respectively. The coefficients of variation obtained from the reproducibility tests were ˂5%. The recoveries of exogenous furosine added to the velvet antler samples were determined at three concentrations (low, intermediate, and high, corresponding to 30, 300, and 3000 ng/mL, respectively). Recovery experiments were performed five times for each concentration, affording values ranging from 93.22 to 95.43%.

### 2.3. Furosine Content in the Processed Velvet Antler

The furosine contents in the different parts of the velvet antler processed by using various methods are shown in Table 1. The furosine contents in the freeze-dried and boiled velvet antler samples were 148.51–193.94 and 168.10–241.22 mg/kg protein, respectively, whereas they were 60.29–80.33 and 115.18–139.88 mg/kg protein for the processed velvet antler without blood and with blood, respectively. These results suggest that the processed velvet antler protein is glycosylated to a considerable extent relative to that in processed foods, such as milk (150–300 mg/kg protein) [30] and processed meat (120 mg/kg protein) [31]. Comparing the contents of furosine, CML, and CEL in different parts of velvet antler processed by various methods [18], we found that the content of furosine in processed velvet antler is between that of CML and CEL. In other words, the CML content is the highest, the furosine content is second, and the CEL content is the lowest in the processed velvet antler.

The furosine contents of the freeze-dried velvet antler were significantly lower than those of the corresponding parts of the boiled samples (*p* < 0.01). This suggests that the processing temperature can significantly affect the formation of furosine. High-temperature processing increased furosine production, whereas the high content in the freeze-dried velvet antler was endogenous. The furosine contents of the velvet antler processed without blood were significantly lower than those of the corresponding parts processed with blood (*p* < 0.01). This may be because the velvet antler processed without blood underwent physical centrifugation to remove the blood. Blood contains many reducing sugars, amino acids, and proteins that can generate furosine during processing. The wax slices exhibited the highest furosine content, whereas the bone slices showed the lowest content. Closer to the top of the antler, more protein, lysine, and reducing sugars were present; these are the precursors of furosine [32,33].

In summary, by comparing the furosine contents in the various parts of velvet antler processed with different methods, we discovered that the furosine contents of freeze-dried velvet antler processed without blood were lower than those of the corresponding parts of boiled velvet antler processed with blood. Wax pieces were more likely to contain furosine than other parts of the velvet antler. Comparing the furosine, CML, and CEL contents in processed velvet antler, we found that the effects of the different processing methods and different parts on the furosine contents in processed velvet antler are similar to those for the CML and CEL contents [18]. Therefore, furosine can be considered as a marker for evaluating the degree of the Maillard reaction and AGE content in processed velvet antler.

### 2.4. Factors Influencing the Furosine Content in the Processed Velvet Antler

The differences in furosine contents between various parts of the velvet antler processed with different methods arise from the occurrence of different degrees of the Maillard reaction. Factors influencing the Maillard reaction include the carbonyl (reducing sugars) and amino (amino acids, proteins) contents, as well as the processing temperature [27,28,34]. Therefore, further investigation of the above-mentioned molecules was performed for different parts of the velvet antler samples processed with the various methods.

#### 2.4.1. Protein and Amino Acid Contents in the Processed Velvet Antler

The protein content of the processed velvet antler samples was determined by using the Dumas combustion method and the related results are shown in Table 2. The protein contents of wax, powder, gauze, and bone slices of the freeze-dried velvet antler were 49.03–81.56%. The protein contents of the boiled velvet antler ranged from 51.13 to 81.25%. For the samples processed without and with blood, the protein contents were 54.11–79.81% and 49.65–82.09%, respectively.

No significant differences (*p* > 0.05) in protein contents were observed between the same parts of the freeze-dried and boiled velvet antler samples. In addition, the parts of the processed velvet antler with blood showed a significantly higher protein content than those processed without blood (*p* < 0.05). This is because the blood (which contains protein) was removed from those samples processed without blood. Significant differences in protein contents between the different parts of the processed velvet antler were observed. The wax slices exhibited significantly higher protein contents than the other parts (*p* < 0.01). This is probably because the wax slices from the antler tip contain an increased amount of meristem tissue, which promotes protein expression [33].

The protein contents of the samples processed with blood are higher than those processed without blood and the protein contents of the wax slices are higher than those of the other parts. Since protein is a substrate for the Maillard reaction, the furosine contents in the samples processed with blood are significantly higher than those in the same parts processed without blood and the furosine contents in the wax slices were significantly higher than those of the other parts.

An automatic amino acid analyzer was used to determine the contents of seventeen amino acids in the various parts of velvet antler processed using different methods. The results of the total amino acid and lysine analyses are summarized in Table 2 and the specific content of all seventeen amino acids is provided in the Appendix A.

The total amino acid and lysine contents in boiled velvet antler were slightly lower than those observed in the same parts of the freeze-dried velvet antler. Heat treatment exacerbates the consumption of amino acids by the Maillard reaction, resulting in significantly higher furosine contents in the boiled velvet antler than those in the corresponding parts of freeze-dried velvet antler. The total amino acid and lysine contents in the samples processed with blood are slightly higher than those processed without blood. The blood in the samples is preserved during the processing and contains a large amount of amino acids, providing a sufficient amount of substrate for the Maillard reaction. Therefore, the furosine contents in the samples processed with blood are significantly higher than those of the corresponding parts processed without blood.

The total amino acid and lysine contents in the various parts of the velvet antler differed significantly. The total amino acid and lysine contents of the wax slices were significantly higher those of the other parts (*p* < 0.01). No significant differences were observed between the powder, gauze, and bone slices (*p* > 0.05). This is probably the result of the wax slices from the antler tip containing more meristem tissue, which increases amino acid demand. Wax slices contain more Maillard reaction substrate amino acids and, therefore, produce more furosine than the other parts.

#### 2.4.2. Reducing Sugar Content in the Processed Velvet Antler

The reducing sugar content of the processed velvet antler was determined by using pre-column-derivatization UPLC. The related chromatogram is shown in Figure 3 and the values are listed in Table 3. Eight monosaccharides were detected in the various parts of velvet antler processed with different methods, namely mannose, glucosamine, ribose, glucuronic acid, galacturonic acid, aminogalactose, glucose, and galactose.

The reducing sugar contents of the freeze-dried velvet antler were significantly higher than those of the corresponding parts of the boiled velvet antler (*p* < 0.05). High temperatures promote the Maillard reaction, so the sugars were consumed to produce furosine. The reducing sugar contents of the samples processed without blood were significantly lower than those of the corresponding parts processed with blood, except for the glucosamine and amino galactose contents (*p* < 0.05). Blood contains large amounts of carbohydrates and its removal by centrifugation decreases the reducing sugar content. Therefore, the samples processed without blood contain less reducing sugar and produce less furosine than the corresponding samples processed with blood.

The wax slices contained significantly higher amounts of reducing sugar than the other parts (*p* < 0.01). The reducing sugar contents in the powder, gauze, and bone slices were gradually reduced *(p* < 0.05). The reason for this is the different amounts of cartilage tissue in the various parts of the velvet antler and the different requirements for sugars [33]. As a consequence of the differential distribution of the Maillard reaction substrates, the furosine content varied in the different parts of the processed velvet antler.

In summary, differences in the furosine content of various parts of velvet antler processed using different methods are caused by the combined effects of reducing sugars, amino acids, proteins, and processing temperature.

## 3. Materials and Methods

### 3.1. Materials

Furosine and TFA were purchased from Sigma–Aldrich (San Francisco, CA, USA). Mannose, glucosamine, ribose, glucuronic acid, galacturonic acid, aminogalactose, glucose, and galactose were purchased from Yuan-ye Biotechnology Co., Ltd. (Shanghai, China). The purity of these reagents was ˃99%. A total of 17 amino acid standards, ninhydrin (NIN), and a citric acid buffer solution were purchased from Hitachi Inc. (Hitachi Co., Osaka, Japan). HPLC-grade acetonitrile was purchased from Fisher Scientific (Waltham, MA, USA), and C18 Sep-Pak^®^ SPE tubes were purchased from Sepax (Sepax technology, Cork, Ireland). Ultrapure water was obtained by using a super-pure water system (Sichuan, China). All other reagents used in this study were of analytical grade and were purchased from Sinopharm Chemical Reagent Co. Ltd. (Beijing, China).

### 3.2. Sources and Preparation of the Velvet Antler

Velvet antler (*Cervi cornu pantotrichum*) was collected in Shuangyang, Jilin Province (China) and was identified by Dr. C.Y. Li from the Chinese Academy of Agricultural Sciences Institute of Special Animal and Plant Sciences.

### 3.3. Preparation of the Different Processed Velvet Antler Slices

According to the classification of commercially available velvet antler, samples that were boiled, freeze-dried, processed with blood, and processed without blood were chosen for this study. A total of six pairs of velvet antler samples were randomly selected to be processed with blood and without blood, for comparison, and another six pairs were randomly selected and processed by boiling and freeze-drying. The boiled velvet antler was boiled for 1 min in boiling water and then baked at a high temperature (75 °C) for 2 h. This operation was repeated several times until dryness was achieved. The freeze-dried velvet antler was directly frozen to dryness. The velvet antler processed without blood was prepared by removing the blood by physical centrifugation, whereas no blood removal was performed for the samples processed with blood. The blood content in velvet antler processed with blood is about 8% by measurement.

Three pairs of boiled and freeze-dried velvet antler samples and three pairs of velvet antler processed with and without blood were randomly selected and crushed whole for subsequent analysis. The remaining six pairs of velvet antler were divided into wax, powder, gauze, and bone slices, on the basis of their morphological and microscopic characteristics (Figure 4) [34]. These parts were segmented, sliced, crushed, sieved, bagged, and labelled for analysis.

### 3.4. Sample Preparation

Pieces of processed velvet antler samples equivalent to 30 mg were mixed with 8 M hydrochloric acid (HCl) and incubated at 110 °C for 24 h. The diluted acid hydrolysate (equivalent to approximately 600 μg of protein) was dried under a nitrogen stream at 70 °C by using a pressured gas-blowing concentrator. The dried hydrolysate was then dissolved in 1 mL of ultra-pure water and solid-phase extracted by using a C18 Sep-Pak^®^ (Sepax Technology, Cork, Ireland) cartridge (500 mg, 6 mL). The SPE column was pre-treated with 3 mL of methanol and 0.1 M TFA at a flow rate of 1 mL/min. The sample was loaded onto the pre-treated SPE column washed with 6 mL of 0.1 M TFA. Finally, the sample was eluted with 3 mL of methanol at a flow rate of 0.5 mL/min. The eluate was dried by freezing, re-dissolved in 1 mL of ultra-pure water, and filtered through a 0.22-μm membrane prior to UPLC-MS/MS analysis.

### 3.5. Quantification of Furosine

The furosine concentration in the hydrolysates was determined by UPLC-MS/MS [35]. Briefly, the protein hydrolysates (2 μg protein, 3 μL) were injected into an Acquity HSS T3 UPLC column (2.1 × 100 mm, 1.8 μm) housed in a column oven at 40 °C and operated in gradient-elution mode. Solvent A was water and solvent B was acetonitrile. Gradient elution began at 80% solvent B for 1 min, followed by a linear gradient from 80% to 30% solvent B over 1.5 min, holding at 30% solvent B for 0.5 min, and then returning to 80% solvent B for 1 min. Analysis was performed by using a Waters Acquity UPLC instrument (Waters, Manchester, UK) coupled to a triple-quadrupole MS operating in MRM mode at a flow rate of 0.3 mL/min. The MS instrument was operated in electrospray-ionization positive mode. The optimized MRM conditions are listed in Table 4. The furosine was quantified by using a pure standard and by reference to an external standard calibration curve. The data are reported as the mean ± standard deviation of triplicate experiments. The furosine content in the samples was expressed as μmol/mmol lysine, μg/g protein, and μg/g sample.

### 3.6. Protein Content Analysis

The protein content of the processed velvet antler was determined by using a Dumas nitrogen analyzer (Velp NDA 701-Monza, Brianza, Italy) according to a previously described method, with minor modifications [29]. The total nitrogen content was converted into the protein content by using a conversion factor of 6.25. The operating conditions of the NDA instrument were as follows: O_2_ gas at 400 mL/min, He gas at 195 mL/min, combustion reactor at 1030 °C, reduction reactor at 650 °C, and pressure of 88.1 kPa.

### 3.7. Amino Acid Content Analysis

An amino acid analyzer (L-8900 System; Hitachi Co., Osaka, Japan) equipped with a visible light detector was used for amino acid analysis. Analytical 2622# (4.6 × 60 mm) and guard 2650# (4.6 × 40 mm) columns were used for the determination of 17 amino acids. Immediately after injection of the sample into the columns, an auto-sampler was used for in-line derivatization by NIN post-column derivatization. NIN-derivatized proline was detected at 440 nm and the other amino acids were detected at 570 nm. Standards of the 17 amino acids were used for identification and quantification (external standard method) and their contents were expressed as g/100 g of processed velvet antler.

### 3.8. Reducing Sugar Content Analysis

The reducing sugars in the processed velvet antler samples were measured with UPLC by using the method reported by Teixeria et al. [36], with minor modifications. The velvet antler samples were boiled with ultra-pure water for 2 h and the reaction was performed with 1 mL of the extract, 0.5 mL of 0.3 M NaOH, and 0.5 mL of 0.3 M 1-phenyl-3-methyl-5-pyrazolone derivatization solution. The reaction mixture was heated in a water bath at 70 °C for 30 min, 0.5 mL of 0.3 M HCl was added to terminate the reaction, and the resultant solution was filtered through a 0.22 μm membrane prior to UPLC analysis. A UPLC instrument equipped with an Acquity UPLC BEH C18 column (100 × 2.1 mm, 1.7 μm) was used. Solvent A was 0.1 M phosphate buffer (pH 7.0) and solvent B was acetonitrile, with a gradient elution of 18% solvent B for 10 min, and a flow rate of 0.3 mL/min. The injection volume was 2 μL and the detection wavelength was 254 nm.

## 4. Conclusions

In the future, analysis of velvet antler and other biological samples from horny drugs, such as nails [37,38], will be a widespread concern. In this study, we established a method for the detection of furosine content in processed velvet antler, which was used for actual sample detection and evaluation of the degree of the Maillard reaction in processed velvet antler. The furosine contents in boiled velvet antler were significantly higher than those in freeze-dried samples. Velvet antler processed with blood exhibited significantly higher levels of furosine than the corresponding parts processed without blood.

Velvet antler boiled at high temperatures produced more furosine than the samples freeze-dried at low temperatures. The contents of Maillard reaction substrates, that is, carbonyl (reducing sugars) and amino (amino acids, proteins) compounds, were determined. The differing contents of these key molecules are the main reason behind the observed differences in furosine contents. For the same parts of processed velvet antler, boiling led to a certain amount of lysine and reducing sugars being consumed to produce more furosine than that in the freeze-dried samples. The samples processed with blood contained more proteins, amino acids, and reducing sugars than those processed without blood, which promoted furosine formation. Similarly, the wax slices contained more proteins, lysine, and reducing sugars and, thus, produced more furosine than the other parts of the velvet antler. The differences in furosine content between the various velvet antler parts, processed by using different methods, were a result of the combined action of the reducing sugars, amino acids, and proteins, as well as the processing temperature.

These data can be used to evaluate the degree of Maillard reaction that has occurred in processed velvet antler samples and can guide the production of low-furosine velvet antler for consumers through optimized processing. In addition, these data can also be used to estimate the furosine intake from velvet antler and educate consumers about how to reduce their furosine intake.

## Figures and Tables

**Figure 1 molecules-24-01255-f001:**
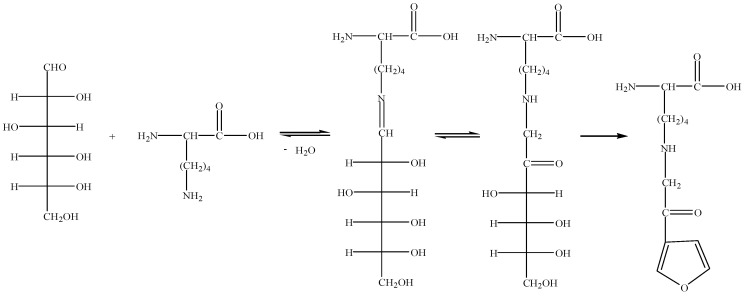
Reaction pathway for conversion of glucose and lysine into furosine via the Maillard reaction.

**Figure 2 molecules-24-01255-f002:**
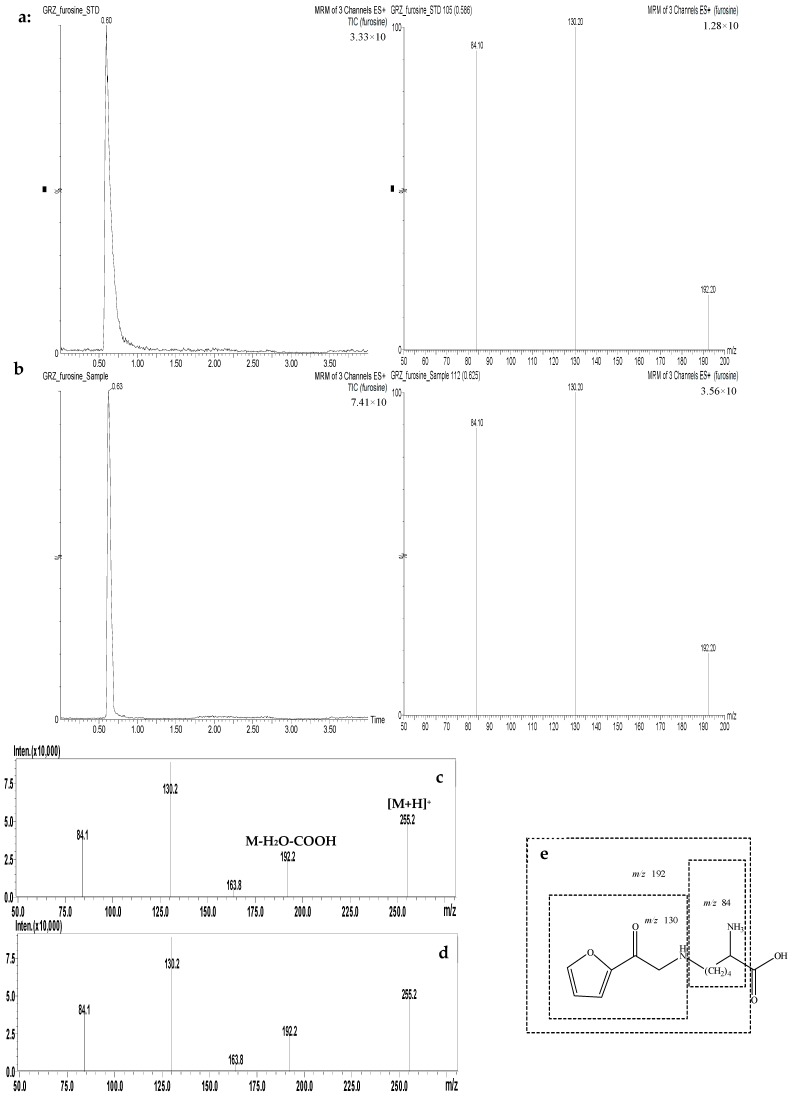
Total ion chromatograms and selected ion intensities for the furosine standard (**a**) and a sample of powder slices of boiled velvet antler (**b**). Mass spectrum fragmentation pattern of the furosine standard (**c**) and a sample of powder slices of boiled velvet antler (**d**). Assignment of the fragmentation pattern of the furosine standard (**e**).

**Figure 3 molecules-24-01255-f003:**
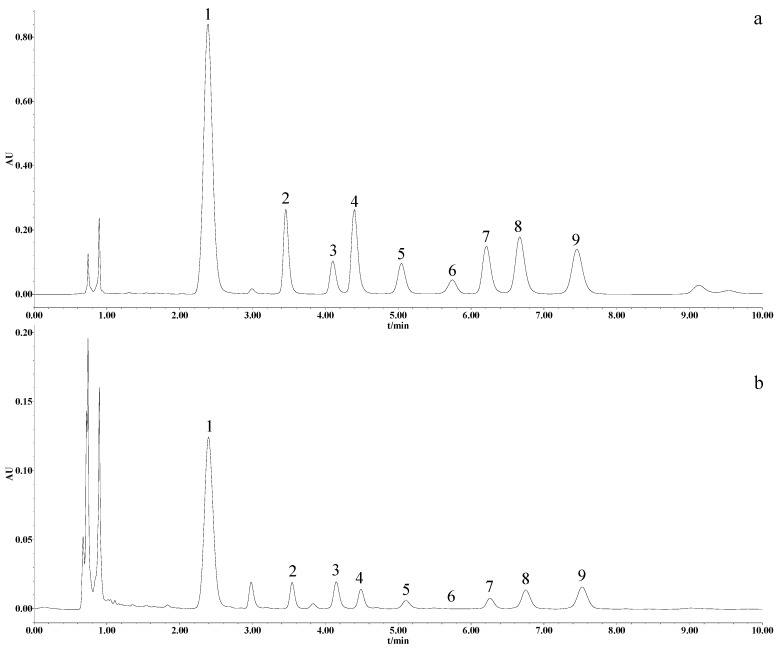
Chromatograms of the eight monosaccharides in (**a**) the mixed standard and (**b**) gauze slices of freeze-dried velvet antler. Peak numbers 1–9 represent 1-phenyl-3-methyl-5-pyrazolone, mannose, glucosamine, ribose, glucuronic acid, galacturonic acid, aminogalactose, glucose, and galactose, respectively.

**Figure 4 molecules-24-01255-f004:**
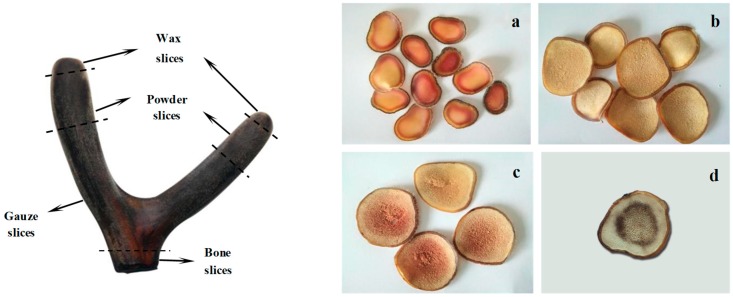
Schematic diagram of different parts of processed velvet antler. The processed velvet antler was divided into wax slices (**a**), powder slices (**b**), gauze slices (**c**), and bone slices (**d**) on the basis of morphological and microscopic characteristics.

**Table 1 molecules-24-01255-t001:** Furosine contents in the processed velvet antler, expressed per mg/kg protein, mg/kg, and mmol/mol lysine ($\bar{x}$ ± SD, *n* = 3).

Processing Methods	Parts	mg FML/kg Protein ^a^	mmol FML/mol Lysine ^b^	mg FML/kg
freeze-dried	wax slices	193.94 ± 1.21	0.53 ± 0.14	138.77 ± 1.78
	powder slices	155.05 ± 1.43	0.50 ± 0.09	88.56 ± 1.84
	gauze slices	154.69 ± 1.19	0.49 ± 0.21	81.58 ± 1.23
	bone slices	153.68 ± 1.98	0.50 ± 0.16	75.35 ± 1.42
	entire	148.51 ± 1.85	0.47 ± 0.20	84.54 ± 1.53
boiled	wax slices	241.22 ± 2.13	0.79 ± 0.23	155.35 ± 1.68
	powder slices	226.21 ± 1.98	0.67 ± 0.22	137.27 ± 1.72
	gauze slices	202.98 ± 1.84	0.62 ± 0.19	109.59 ± 1.66
	bone slices	155.02 ± 1.78	0.57 ± 0.24	79.26 ± 2.03
	entire	168.10 ± 1.22	0.56 ± 0.26	102.54 ± 1.22
processed without blood	wax slices	80.33 ± 1.25	0.26 ± 0.12	64.11 ± 2.04
	powder slices	61.65 ± 1.31	0.20 ± 0.14	34.96 ± 1.75
	gauze slices	49.26 ± 1.43	0.17 ± 0.21	29.01 ± 2.10
	bone slices	50.35 ± 1.13	0.17 ± 0.19	27.24 ± 1.45
	entire	60.29 ± 1.46	0.25 ± 0.17	34.02 ± 1.58
processed with blood	wax slices	139.88 ± 1.87	0.44 ± 0.22	106.62 ± 1.95
	powder slices	126.42 ± 1.54	0.42 ± 0.19	83.89 ± 1.63
	gauze slices	122.46 ± 1.44	0.41 ± 0.25	75.25 ± 1.39
	bone slices	129.15 ± 1.32	0.35 ± 0.17	64.13 ± 1.88
	entire	115.18 ± 2.13	0.38 ± 0.14	68.24 ± 1.21

^a^ Data were calculated using the protein contents quantified by combustion method. ^b^ Data were calculated using the amino acid concentration in the acid hydrolysates, quantified by an amino acid analyzer.

**Table 2 molecules-24-01255-t002:** Protein, total amino acids, and lysine contents in the processed velvet antler ($\bar{x}$ ± SD, *n* = 3).

Processing Methods	Parts	Protein/%	Total Amino Acids/g/100 g	Lysine/g/100 g
freeze-dried	wax slices	81.56 ± 0.04	88.64 ± 3.30	5.87 ± 0.20
	powder slices	57.12 ± 0.03	62.51 ± 1.84	3.96 ± 0.11
	gauze slices	52.74 ± 0.10	61.75 ± 0.90	3.76 ± 0.03
	bone slices	49.03 ± 0.25	58.20 ± 1.17	3.41 ± 0.21
	entire	56.93 ± 0.34	68.00 ± 1.23	4.02 ± 0.13
boiled	wax slices	81.25 ± 0.12	87.48 ± 1.78	5.67 ± 0.22
	powder slices	58.11 ± 0.18	62.29 ± 0.82	3.92 ± 0.14
	gauze slices	53.99 ± 0.33	60.97 ± 0.28	3.69 ± 0.10
	bone slices	51.13 ± 0.25	55.62 ± 1.74	3.15 ± 0.12
	entire	60.99 ± 0.44	67.59 ± 1.82	4.09 ± 0.29
processed without blood	wax slices	79.81 ± 0.09	86.74 ± 0.18	5.56 ± 0.11
	powder slices	56.69 ± 0.11	66.65 ± 0.74	4.01 ± 0.09
	gauze slices	58.88 ± 0.31	62.35 ± 0.48	3.76 ± 0.11
	bone slices	54.11 ± 0.24	52.34 ± 0.72	3.01 ± 0.12
	entire	56.43 ± 0.28	63.55 ± 0.82	3.58 ± 0.15
processed with blood	wax slices	82.09 ± 0.74	91.02 ± 0.46	5.66 ± 0.12
	powder slices	61.49 ± 0.33	74.06 ± 0.34	4.81 ± 0.02
	gauze slices	61.45 ± 0.41	66.94 ± 0.49	4.26 ± 0.07
	bone slices	49.65 ± 0.56	61.06 ± 0.46	3.48 ± 0.15
	entire	59.25 ± 0.35	64.31 ± 0.56	4.03 ± 0.17

**Table 3 molecules-24-01255-t003:** Eight monosaccharides contents in processed velvet antler, expressed per mg/kg ($\bar{x}$ ± SD, *n* = 3).

**Compound**	**Freeze-Dried**				**Boiled**			
	**Wax Slices**	**Powder Slices**	**Gauze Slices**	**Bone Slices**	**Wax Slices**	**Powder Slices**	**Gauze Slices**	**Bone Slices**
mannose	13.31 ± 1.01	9.49 ± 0.78	9.75 ± 0.65	7.66 ± 0.55	10.10 ± 0.98	6.32 ± 0.42	5.86 ± 0.45	4.31 ± 0.33
glucosamine	70.21 ± 2.03	60.32 ± 1.84	58.45 ± 1.21	48.04 ± 1.97	66.34 ± 2.33	56.30 ± 1.04	51.75 ± 1.32	44.04 ± 1.74
ribose	14.16 ± 1.21	9.37 ± 0.96	7.94 ± 1.01	6.60 ± 0.95	7.63 ± 1.04	5.52 ± 0.97	4.07 ± 0.87	3.58 ± 0.74
glucuronic acid	20.02 ± 1.19	16.51 ± 1.20	12.09 ± 0.98	7.46 ± 0.77	15.34 ± 1.04	13.74 ± 0.96	8.05 ± 0.81	6.18 ± 0.77
galacturonic acid	3.55 ± 0.94	2.64 ± 0.81	1.26 ± 0.72	0.27 ± 0.18	2.75 ± 0.55	1.52 ± 0.53	1.29 ± 0.19	0.20 ± 0.12
aminogalactose	18.36 ± 1.32	15.77 ± 1.09	11.39 ± 0.75	3.96 ± 0.63	16.98 ± 0.94	11.32 ± 1.02	9.88 ± 0.91	2.39 ± 0.74
glucose	18.10 ± 1.52	13.46 ± 1.04	11.95 ± 1.22	10.13 ± 1.09	16.25 ± 1.43	10.86 ± 1.51	9.24 ± 1.06	7.13 ± 1.33
galactose	28.89 ± 1.82	26.54 ± 2.07	26.58 ± 1.67	22.33 ± 1.33	22.93 ± 1.47	16.62 ± 1.83	16.75 ± 1.42	19.43 ± 1.17
**Compound**	**Processed with Blood**				**Processed without Blood**			
	**Wax Slices**	**Powder Slices**	**Gauze Slices**	**Bone Slices**	**Wax Slices**	**Powder Slices**	**Gauze Slices**	**Bone Slices**
mannose	11.20 ± 1.07	9.60 ± 1.21	6.30 ± 0.97	5.83 ± 0.84	9.22 ± 1.07	7.15 ± 0.92	4.77 ± 0.67	3.22 ± 0.54
glucosamine	52.34 ± 2.07	35.47 ± 2.22	31.34 ± 1.98	28.10 ± 2.13	57.36 ± 2.45	38.11 ± 1.56	33.71 ± 1.42	30.13 ± 1.08
ribose	5.63 ± 0.71	4.78 ± 0.63	3.22 ± 0.58	2.10 ± 0.41	4.17 ± 0.51	3.90 ± 0.35	2.48 ± 0.24	1.16 ± 0.16
glucuronic acid	15.14 ± 0.74	13.83 ± 0.81	11.62 ± 0.53	5.01 ± 0.37	13.55 ± 1.02	8.76 ± 0.91	7.01 ± 0.87	3.65 ± 0.54
galacturonic acid	2.55 ± 0.32	1.90 ± 0.28	1.62 ± 0.17	0.82 ± 0.19	2.19 ± 0.23	1.72 ± 0.31	1.60 ± 0.22	0.20 ± 0.12
aminogalactose	16.78 ± 0.79	12.92 ± 0.82	9.46 ± 0.77	7.06 ± 0.54	20.24 ± 0.86	16.77 ± 0.72	12.34 ± 0.83	0.57 ± 0.14
glucose	16.25 ± 0.44	12.68 ± 0.58	9.21 ± 0.79	7.38 ± 0.88	13.65 ± 0.92	9.50 ± 0.77	7.62 ± 0.81	5.91 ± 0.69
galactose	22.73 ± 1.07	18.23 ± 0.98	16.44 ± 1.21	13.07 ± 0.78	18.08 ± 1.23	17.59 ± 1.42	15.22 ± 1.33	12.04 ± 0.79

**Table 4 molecules-24-01255-t004:** UPLC-MS settings for multiple reaction monitoring (MRM).

Compound	Precursor Ion (*m*/*z*)	Product Ion (*m*/*z*)	Cone Voltage (V)	Collision Energy (ev)	Dwell Time (ms)
furosine	255	192	25	15	36
	255	130	25	15	36
	255	84	25	25	36

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
