# Peer review of "Quantification of Furosine (Nε-(2-Furoylmethyl)-l-lysine) in Different Parts of Velvet Antler with Various Processing Methods and Factors Affecting Its Formation"

_molecules, 2019, doi:10.3390/molecules24071255_

Reviewer 1 Report

Molecules

Review of Manuscript Number: molecules-466705

TITLE:  Quantification of furosine (Nε-(2-furoylmethyl)-L-lysine) in different parts of antler velvet with various processing methods and factors affecting its formation

CORRESPONDING AUTHOR: Yin-shi Sun

General Comments

This manuscript addresses the development and evaluation of determination of furosine for food analysis in antler velvet. The reverse-phase column (HSS T3) HPLC column was used to analyze for actual antler velvet samples with sensitivity, the possibility of the results of determination of furosine was expected for immunity or pharmacology in the future. Although, I found some of the authors' explanations difficult to follow; I suspect a reader less familiar with the topic might have even greater difficulties. I would suggest major revision in combination with re-review for this manuscript. I annotate the manuscript with several corrections listed below, which I believe will improve the readability of the paper.

1: Major comments

(1)          The authors discussed content of furosine for evaluation of antler velvet, herein, the analytical application was well carried out by UPLC-MS. The determination results of furosine are interesting in this study. However, I checked author’s previous article (reference 18), the style of this manuscript construction is well similar. I suggest that the authors should insert the comparative discussion between previous study (CML and CEL) and this study (furosine) in the manuscript.

Alternatively, the article could be moved into a short communication waiting to test other food or pharmacological substances. The article can be accepted if the aforementioned requests are respected.

(2)          The authors used the UPLC-MS/MS technique as effective determination method for furosine. However, furosine could be determined by UV absorption measurement, which was described in Introduction Section (Line 53-54). To respect this description and phenomenon, the validation of proposed method should be carried out. I suggest that the addition of description for including the comparative study of sensitivity or selectivity to UV measurements in Introduction Section or Results and Discussion section.

(3)          HSS T3 UPLC column was used for the effective determination of furosine in this manuscript. In contrast, HILIC column have been also used for selective determination of high polar chemical species. The authors should discuss the reason of the use of HSS T3 column compared to HILIC column in main text. In addition, please insert some discussion including an interaction between HSS T3 stationary phase and furosine.

(4)          The authors concluded that the concentration of precursor chemical species was increased from bone sample to wax sample (root to top). In generally, it seemed that the amount of precursor chemical species is decreased with exploring natural condition (including wash, rinse and degradation). Please insert some reasons to concentrate for the wax samples.

Furthermore, reference No. 32 was not supported the reason or results of concentration of precursors in antler (or skin). Could you please check the manuscript and overall references form again?

(5)          I could not find Figure 3 in manuscript.

2: Minor comments

(1)          The abbreviations (CML and CEL, Line 43; FML, Table 1) were not defined in manuscript.

(2)          m/z is amount of substance. Could you please correct to italicize Figure 2 and Table 4.

(3)          Could you please insert the explanation of LOD and LOQ derivation from the linear range (Line 99-102). At first glance, it seemed that the LOQ value was left from the determination range.

(4)          The authors concluded that the blood was one of the important roles of the proposed determination method of furosine from Table 1-4. I thought that the concentration (content) of blood in samples were important for the accurate determination. I suggest that the authors insert the results of concentration of blood in samples in main text.

(5)          I can not follow the mean of “entire” in Table 1. Please explain this word in detail.

(6)          The spell of “wax” is mistaken in Figure 4.

(7)          The chromatograms at 440 nm in Supporting information (S1, b) is not cleared. Please correct this chromatogram.

I hope that my comment is useful for the improvement of the article.

Author Response

Dear reviewer,

We must thank you for your valuable comments and thoughtful suggestions. These valuable comments not only helped us with the improvement of our manuscript, but also suggested some neat ideas for our future studies. Please do forward our heartfelt thanks to these experts.

Based on the comments we received, careful modifications have been made to the whole manuscript. All changes made to the text were summarized and were clearly marked in red. In addition, we also consulted a native English speaker for paper revision before submission. We hope the new manuscript will meet your magazine’s standard. Below you will find our point-by-point responses to the editor and the reviewers’ comments. They were summarized in 2 separate documents: 1. Response to reviewer 1’s comments; 2. Other changes.

If you need further information, please do not hesitate to contact me.

Yours sincerely,

Yinshi Sun

Document 1. Responses to reviewer 1’s comments

1: Major comments

(1) The authors discussed content of furosine for evaluation of antler velvet, herein, the analytical application was well carried out by UPLC-MS. The determination results of furosine are interesting in this study. However, I checked author’s previous article (reference 18), the style of this manuscript construction is well similar. I suggest that the authors should insert the comparative discussion between previous study (CML and CEL) and this study (furosine) in the manuscript.Alternatively, the article could be moved into a short communication waiting to test other food or pharmacological substances. The article can be accepted if the aforementioned requests are respected.

Response: Thank you for your valuable comments and thoughtful suggestions. We have inserted the comparative discussion between our previous study (CML and CEL) and this study (furosine) in the revised manuscript according to your suggestion (Line 123-126 and Line 144-148). Comparing the contents of furosine, CML and CEL in different parts of velvet antler with different processing methods, we found that the CML content of processed velvet antler is the highest, the furosine content is second, and the CEL content is the lowest. Comparing the furosine, CML and CEL in processed velvet antler, we found that the effect of different processing methods and different parts on furosine content in processed velvet antler is similar to that of CML and CEL.

(2) The authors used the UPLC-MS/MS technique as effective determination method for furosine. However, furosine could be determined by UV absorption measurement, which was described in Introduction Section (Line 53-54). To respect this description and phenomenon, the validation of proposed method should be carried out. I suggest that the addition of description for including the comparative study of sensitivity or selectivity to UV measurements in Introduction Section or Results and Discussion section.

Response: We are grateful to the reviewer for pointing out our defect. We have added the comparative of sensitivity and selectivity between LC-MS and UV measurements in Introduction Section (Line 55-56). Furosine is a trace substance in foods and drugs, LC-MS has better sensitivity and selectivity than UVmeasurements. Therefore, LC-MS is usually used to detect furosine in foods and drugs at now.

(3) HSS T3 UPLC column was used for the effective determination of furosine in this manuscript. In contrast, HILIC column have been also used for selective determination of high polar chemical species. The authors should discuss the reason of the use of HSS T3 column compared to HILIC column in main text. In addition, please insert some discussion including an interaction between HSS T3 stationary phase and furosine.

Response: We are grateful to the reviewer for pointing out our defect. We have discussed the reason of the use of HSS T3 column compared to HILIC column, compared to the HILIC column, the HSS T3 column stationary phase is compatible with 100% water and has a wider elution range (Line 85-86). In addition, The HSS T3 column relies on ion exchange and hydrophobic interaction between the stationary phase and furosine to achieve separation (Line 84-85).

(4) The authors concluded that the concentration of precursor chemical species was increased from bone sample to wax sample (root to top). In generally, it seemed that the amount of precursor chemical species is decreased with exploring natural condition (including wash, rinse and degradation). Please insert some reasons to concentrate for the wax samples. Furthermore, reference No. 32 was not supported the reason or results of concentration of precursors in antler (or skin). Could you please check the manuscript and overall references form again?

Response: We are very sorry that our explanation is not enough to make it difficult for you to understand. As you said, the amount of precursor chemical species is decreased with exploring natural condition (including wash, rinse and degradation). However, the structure of the velvet antler is very unique. The top of the wax slice has many meristems, which are constantly differentiated and grow, so it has more amino acids, proteins and reducing sugars. The bottom(bone slice) of the tissue is gradually calcified, aging and the above substances are gradually reduced.

The purpose of reference 32 is to show that amino acids, proteins and reducing sugars are precursors of furosine. Reference 33 and subsequent experiments were able to support closer to the top of the antler, more protein, lysine, and reducing sugars were present(Line 167-169, 190-192 and 211-215). In addition, we checked all references to ensure proper citations in the revised manuscript.

(5)  I could not find Figure 3 in manuscript.

Response: We are sorry that Figure 3 was lost on upload and we have re-added it in the revised manuscript (behind Line 198).

2: Minor comments

(1) The abbreviations (CML and CEL, Line 43; FML, Table 1) were not defined in manuscript.

Response: We are grateful to the reviewer for pointing out our defect, we have defined abbreviations for CML, CEL and FML in the revised manuscript (Line 42-43 and 49-50).

(2) m/z is amount of substance. Could you please correct to italicize Figure 2 and Table 4.

Response: Thank you for your pointing out our defect, We have corrected the m/z in Figure 2 and Table 4.

(3) Could you please insert the explanation of LOD and LOQ derivation from the linear range (Line 99-102). At first glance, it seemed that the LOQ value was left from the determination range.

Response: We are grateful to the reviewer for pointing out our defect, we have inserted the explanation of LOD and LOQ derivation in the revised manuscript (Line 105-108).

(4) The authors concluded that the blood was one of the important roles of the proposed determination method of furosine from Table 1-4. I thought that the concentration (content) of blood in samples were important for the accurate determination. I suggest that the authors insert the results of concentration of blood in samples in main text.

Response: Thank you for your valuable comments and thoughtful suggestions. Of course, the content of blood in processed velvet antler with blood samples were important for the accurate determination. Therefore, we have inserted the results of concentration of blood in processed velvet antler with blood samples in the revised manuscript (Line 242-243). The blood content in processed velvet antler withblood is about 8% by measurement.

(5) I can not follow the mean of “entire” in Table 1. Please explain this word in detail.

Response: We are very sorry that our describe make it difficult for you to understand. The word “ entire” in Table 1 means the processed velvet antler crushed whole for subsequent analysis. This part of the content we have explained in Section 3.3 (Line 244- 248).

(6) The spell of “wax” is mistaken in Figure 4.

Response: Thank you for your pointing out our mistake, We have corrected the spell of wax in Figure 4.

(7) The chromatograms at 440 nm in Supporting information (S1, b) is not cleared. Please correct this chromatogram. 

Response: Thank you for your suggestion. We resubmitted a clear chromatogram (S1, b) at 440 nm in the supplemental material.

Document 2. Other changes

1. To address the reviewers’ comments regarding English language and style, we have worked to improve the flow, clarity, and readability of the English in the manuscript. We have edited the manuscript for grammar, spelling, punctuation, and syntax. We have checked the consistency and style of the manuscript.

2. We have placed the keywords in alphabetical order, as this is preferred by most journals; We replaced the “processing metnod of antler velvet” with “velvet antler processing”, Shorter keywords are usually preferred by journals and have more impact.

3. Sentence amended to improve readability and clarity, such as “The relative standard deviations of the intra-day and inter-day precision for furosine were 3.12 and 4.28%, respectively”(Line 110-111).

Reviewer 2 Report

Here, below, there is the review of Manuscript (molecules-466705) entitled “Quantification of furosine (Nε-(2-furoylmethyl)-L-lysine) in different parts of antler velvet with various processing methods and factors affecting its formation”

Generally, creation of Amadori products during Amadori rearrangement which is connected with Maillard reaction fulfils the crucial role in food chemistry. So the various researches connected with this subject are important and justified. As a matter of fact, the velvet antler is rather uncommon as food in west countries but it has group of consumers. Below, there are my remarks concerning directly the manuscript:

1. “Velvet antler” phrase should be rather used than “antler velvet” in the title and in the text of the manuscript. Please check it out.

2. Systematic name of furosine should be given in accordance with IUPAC recommendation.

3. Abbreviations CML and CEL should be clearly explained. Generally, for clarity, all abbreviations should be given with their descriptions at the beginning of manuscript.

4. Fig 2 encloses only MS spectrum of reference furosine. It is necessary to add MS spectrum of furosine, which was got from the tested sample. Lack of relative intensities of molecular and fragment ion peaks. Calculated isotope mass of furosine should be added. Positive ESI-MS gives quasi molecular ion peaks because of addition of proton or other positive charged moiety to considered molecule what also depends on solvent used to measuring of ESI-MS spectrum. Analysis of isotope peaks of quasi molecular ion is suggested.

5. Determination of structure of potential furosine extracted from processed velvet antler is insufficient. Additional tests should be conducted that prove that a compound, extracted from the tested samples of the velvet antlers, to be furosine. For example, IR spectrum should be recorded, and then compared with IR of reference furosine. Also NMR spectra are recommended and especially 13C-NMR is suggested.

6. In tables 1 and 2, for more clarity, the information of average value (mean), standard deviation and number of samples (n) is suggested to be added as it was done for in the case of table 3.

7. Some references are prepared incorrectly. The Authors of this manuscript frequently mix up first names with surnames, for example, ref 13, 15, 23, 27 and 32. It is necessary to check out all references. According to journal’s requirements the reference should be cited author’s surname that is followed with the initial of author’s first name.

Author Response

Dear reviewer,

We must thank you for your valuable comments and thoughtful suggestions. These valuable comments not only helped us with the improvement of our manuscript, but also suggested some neat ideas for our future studies. Please do forward our heartfelt thanks to these experts.

Based on the comments we received, careful modifications have been made to the whole manuscript. All changes made to the text were summarized and were clearly marked in red. In addition, we also consulted a native English speaker for paper revision before submission. We hope the new manuscript will meet your magazine’s standard. Below you will find our point-by-point responses to the editor and the reviewers’ comments. They were summarized in 3 separate documents: 1. Response to reviewer 2’s comments; 2. Other changes.

If you need further information, please do not hesitate to contact me.

Yours sincerely,

Yinshi Sun

Document 1. Responses to reviewer 2’s comments

1. “Velvet antler” phrase should be rather used than “antler velvet” in the title and in the text of the manuscript. Please check it out.

Response: Thank you for your valuable comments and thoughtful suggestions. We have replaced “antler velvet” with “velvet antler” in the revised manuscript .

2. Systematic name of furosine should be given in accordance with IUPAC recommendation.

Response: Thank you for your suggestion. Furosine and Nε-(2-furoylmethyl)-L-lysine are common names that people often used, and the systematic name of furosine N6-[2-(2-furanyl)-2-oxoethyl]-L

-lysine has been added the revised manuscript (Line 49-50).

3. Abbreviations CML and CEL should be clearly explained. Generally, for clarity, all abbreviations should be given with their descriptions at the beginning of manuscript.

Response: We are grateful to the reviewer for pointing out our defect, we have defined abbreviations for CML, CEL and FML in the revised manuscript (Line 42-43 and 49-50).

4. Fig 2 encloses only MS spectrum of reference furosine. It is necessary to add MS spectrum of furosine, which was got from the tested sample. Lack of relative intensities of molecular and fragment ion peaks. Calculated isotope mass of furosine should be added. Positive ESI-MS gives quasi molecular ion peaks because of addition of proton or other positive charged moiety to considered molecule what also depends on solvent used to measuring of ESI-MS spectrum. Analysis of isotope peaks of quasi molecular ion is suggested.

Response: Thank you for your pointing out our defect. We have added Mass spectrum fragmentation pattern of powder slices of boiled velvet antler samples in the revised manuscript (Fig 2-d). The relative intensities of molecular and fragment ion peaks of furosine standard (Fig 2-a) and powder slices of boiled velvet antler samples ( Fig 2-b) also were added in the revised manuscript. Calculated isotope mass of furosine (C12H18N2O4, Mw 254.12) was added in the revised manuscript (Line 49-50). We have given the positive ESI-MS quasi molecular ion peaks of furosine (Fig 2-c) in the revised manuscript. In the early ion optimization, we excluded [M+Na]+ and [M+K]+ in consideration of the sensitivity, and finally selected [M+H]+ as the excimer ion peak. In Figure 2-c and Figure 2-e, we analyzed and attributed the fragment peak of furosine.

5. Determination of structure of potential furosine extracted from processed velvet antler is insufficient. Additional tests should be conducted that prove that a compound, extracted from the tested samples of the velvet antlers, to be furosine. For example, IR spectrum should be recorded, and then compared with IR of reference furosine. Also NMR spectra are recommended and especially 13C-NMR is suggested.

Response: Thank you for your suggestion. The key objectives of this study was performed to determine the furosine content in processed velvet antler by UPLC-MS/MS and to explore the factors affecting its content. Determination of the structure of furosine is not the main content of this study. But we performed additional tests, such as UV (Figure 1) same as Lee’s report [35], to demonstrate that the compound extracted from the test sample of the velvet antler was furosine. We can provide a UV determination of furosine standards and powder slices of boiled velvet antler samples as shown in Figure 1 below. We considered that the same retention time, UV measurement and UPLC-MS/MS can fully demonstrate the furosine in the velvet antler samples.

6. In tables 1 and 2, for more clarity, the information of average value (mean), standard deviation and number of samples (n) is suggested to be added as it was done for in the case of table 3.

Response: Thank you for your valuable comments and thoughtful suggestions. We have added the information of average value (mean), standard deviation and number of samples (n) in tables 1 and 2 in the revised manuscript.

7. Some references are prepared incorrectly. The Authors of this manuscript frequently mix up first names with surnames, for example, ref 13, 15, 23, 27 and 32. It is necessary to check out all references. According to journal’s requirements the reference should be cited author’s surname that is followed with the initial of author’s first name.

Response: We are very sorry that the format of the reference has been incorrect. In the revised manuscript, we checked the format of the reference carefully to ensure it was properly quoted.

Document 2. Other changes

1. To address the reviewers’ comments regarding English language and style, we have worked to improve the flow, clarity, and readability of the English in the manuscript. We have edited the manuscript for grammar, spelling, punctuation, and syntax. We have checked the consistency and style of the manuscript.

2. We have placed the keywords in alphabetical order, as this is preferred by most journals; We replaced the “processing metnod of antler velvet” with “velvet antler processing”, Shorter keywords are usually preferred by journals and have more impact.

3. Sentence amended to improve readability and clarity, such as “The relative standard deviations of the intra-day and inter-day precision for furosine were 3.12 and 4.28%, respectively”(Line 110-111).

Round  2

Reviewer 1 Report

Molecules

Review of Manuscript Number: molecules-466705

TITLE:  Quantification of furosine (Nε-(2-furoylmethyl)-L-lysine) in different parts of velvet antler with various processing methods and factors affecting its formation

CORRESPONDING AUTHOR: Yin-shi Sun

General Comments

The authors have substantially revised their paper and thus dramatically improved its quality. The author’s replies to reviewer comments are detailed and convincing, and show that they have taken seriously the reviewer’s advice. I propose that the manuscript can be acceptable to Molecules, with following minor comments.

Minor comments

1: This study proposed an analytical method of furosine in velvet antler. In review process, I received an interesting information that an interaction between chemical species and keratinous types. The results including pretreatment will provide a useful proposal for not only food chemistry area but also pharmacological chemistry area. Especially, analytical results of drugs from keratinous biological samples have been essentially interested for readers; such as hair or nail analysis. If possible, I suggest that a simple one sentence (future prospects) should be add in conclusions section with following references as examples;

(1) Renata Solimini, Adele Minutillo, Chrystalla Kyriakou, Simona Pichini, Roberta Pacifici, Francesco Paolo Busardo. Nails in Forensic Toxicology: An Update. Curr. Pharm. Des. 2017;23(36):5468-5479. doi: 10.2174/1381612823666170704123126.

(2) Francesco Paolo Busardò, Massimo Gottardi, Anastasio Tini, Claudia Mortali, Raffaele Giorgetti, Simona Pichini. Ultra-High-Performance Liquid Chromatography Tandem Mass Spectrometry Assay for Determination of Endogenous GHB and GHB-Glucuronide in Nails. Molecules 2018, 23(10), 2686; https://doi.org/10.3390/molecules23102686

(3) Fumiki Takahashi, Masaru Kobayashi, Atsushi Kobayashi, Kanya Kobayashi, Hideki Asamura. High-Frequency Heating Extraction Method for Sensitive Drug Analysis in Human Nails. Molecules 2018, 23(12), 3231; https://doi.org/10.3390/molecules23123231

2: The labels and units in Figure 3 are not clear. Could the authors please increase fonts.

I hope that my comments are useful for the improvement of this article.

Author Response

Dear reviewer,

We must thank you for your valuable comments and thoughtful suggestions. These valuable comments not only helped us with the improvement of our manuscript, but also suggested some neat ideas for our future studies. Please do forward our heartfelt thanks to these experts.

Based on the comments we received and previous revisions, we have revised the whole manuscript carefully. All changes made to the text were summarized and were clearly marked in red. In addition, we also consulted a native English speaker for paper revision before submission. We hope the new manuscript will meet your magazine’s standard. Below you will find our point-by-point responses to the editor and the reviewers’ comments. They were summarized in 3 separate documents: 1. Response to reviewer 1’s comments; 2. Other changes.

If you need further information, please do not hesitate to contact me.

Yours sincerely,

Yinshi Sun

Document 1. Responses to reviewer 1’s comments

Minor comments

1:This study proposed an analytical method of furosine in velvet antler. In review process, I received an interesting information that an interaction between chemical species and keratinous types. The results including pretreatment will provide a useful proposal for not only food chemistry area but also pharmacological chemistry area. Especially, analytical results of drugs from keratinous biological samples have been essentially interested for readers; such as hair or nail analysis. If possible, I suggest that a simple one sentence (future prospects) should be add in conclusions section with following references as examples;

(1) Renata Solimini, Adele Minutillo, Chrystalla Kyriakou, Simona Pichini, Roberta Pacifici, Francesco Paolo Busardo. Nails in Forensic Toxicology: An Update. Curr. Pharm. Des. 2017;23(36):5468-5479. doi: 10.2174/1381612823666170704123126.

(2) Francesco Paolo Busardò, Massimo Gottardi, Anastasio Tini, Claudia Mortali, Raffaele Giorgetti, Simona Pichini. Ultra-High-Performance Liquid Chromatography Tandem Mass Spectrometry Assay for Determination of Endogenous GHB and GHB-Glucuronide in Nails. Molecules 2018, 23(10), 2686; https://doi.org/10.3390/molecules23102686

(3) Fumiki Takahashi, Masaru Kobayashi, Atsushi Kobayashi, Kanya Kobayashi, Hideki Asamura. High-Frequency Heating Extraction Method for Sensitive Drug Analysis in Human Nails. Molecules 2018, 23(12), 3231; https://doi.org/10.3390/molecules23123231

Response: Thank you for your valuable comments and thoughtful suggestions. Of course, the results of analysis of drugs from keratinous biological samples are of increasing interest; such as hair or nail analysis. Therefore, we added a simple future prospects in conclusions section follow the comments of reviewers (Line 302-303). At the same time, we quote references (No. 38 and 39)recommended by reviewers.

2: The labels and units in Figure 3 are not clear. Could the authors please increase fonts.

Response: We are grateful to the reviewer for pointing out our defect. We re-uploaded a new Figure 3 with clear units and labels. We have increased the font of the units and labels.

Document 2. Other changes

1. At the end of the manuscript we added sample availability.

Reviewer 2 Report

Comments after correction of Manuscript (molecules-466705) entitled “Quantification of furosine (Nε-(2-furoylmethyl)-L-lysine) in different parts of antler velvet with various processing methods and factors affecting its formation”

1. Please, insert the sent figure 1, which shows comparison of UV spectra of reference furosine and the extracted one from tested sample, in the manuscript or at least in the supplementary material. It is necessary to realize that these studies would be pointless without 100% certainty that the extracted substance from tested samples is clean furosine. That is why the certain proofs of the unambiguous determination of the chemical structure of the isolated compound are so important.

2. A format of some references (e.g. 15, 27 and 32) is still incorrect. The Authors still mix up first name with surname or give incorrect surnames in cited references. Please, check them out carefully and correct them.

Author Response

Dear reviewer,

We must thank you for your valuable comments and thoughtful suggestions. These valuable comments not only helped us with the improvement of our manuscript, but also suggested some neat ideas for our future studies. Please do forward our heartfelt thanks to these experts.

Based on the comments we received and previous revisions, we have revised the whole manuscript carefully. All changes made to the text were summarized and were clearly marked in red. In addition, we also consulted a native English speaker for paper revision before submission. We hope the new manuscript will meet your magazine’s standard. Below you will find our point-by-point responses to the editor and the reviewers’ comments. They were summarized in 3 separate documents: 1. Response to reviewer 2’s comments; 2. Other changes.

If you need further information, please do not hesitate to contact me.

Yours sincerely,

Yinshi Sun

Document 1. Responses to reviewer 2’s comments

1. Please, insert the sent figure 1, which shows comparison of UV spectra of reference furosine and the extracted one from tested sample, in the manuscript or at least in the supplementary material. It is necessary to realize that these studies would be pointless without 100% certainty that the extracted substance from tested samples is clean furosine. That is why the certain proofs of the unambiguous determination of the chemical structure of the isolated compound are so important.

Response: Thank you for your valuable comments and thoughtful suggestions. Of course, the certain proofs of the unambiguous determination of the chemical structure of the isolated compound are important, such as UV spectra. Therefore, we introduced this part in the text (Line 92-95) and uploaded the UV chromatogram of furosine standards and powder slices of boiled velvet antler samples in the supplementary material (Figure 1) according to the reviewer's comments.

2. A format of some references (e.g. 15, 27 and 32) is still incorrect. The Authors still mix up first name with surname or give incorrect surnames in cited references. Please, check them out carefully and correct them.

Response: We are very sorry that the format of the reference has been incorrect. In the revised manuscript, we checked the format of the reference (e.g. 15, 27 and 32) carefully to ensure it was properly quoted.

Document 2. Other changes

1. At the end of the manuscript we added sample availability.
